# Conditional Image Generation with PixelCNN Decoders

**Aäron van den Oord**
Google DeepMind
avdnoord@google.com

**Nal Kalchbrenner**
Google DeepMind
nalk@google.com

**Oriol Vinyals**
Google DeepMind
vinyals@google.com

**Lasse Espeholt**
Google DeepMind
espeholt@google.com

**Alex Graves**
Google DeepMind
gravesa@google.com

**Koray Kavukcuoglu**
Google DeepMind
korayk@google.com

## Abstract

This work explores conditional image generation with a new image density model based on the PixelCNN architecture. The model can be conditioned on any vector, including descriptive labels or tags, or latent embeddings created by other networks. When conditioned on class labels from the ImageNet database, the model is able to generate diverse, realistic scenes representing distinct animals, objects, landscapes and structures. When conditioned on an embedding produced by a convolutional network given a single image of an unseen face, it generates a variety of new portraits of the same person with different facial expressions, poses and lighting conditions. We also show that conditional PixelCNN can serve as a powerful decoder in an image autoencoder. Additionally, the gated convolutional layers in the proposed model improve the log-likelihood of PixelCNN to match the state-of-the-art performance of PixelRNN on ImageNet, with greatly reduced computational cost.

## 1 Introduction

Recent advances in image modelling with neural networks [30, 26, 20, 10, 9, 28, 6] have made it feasible to generate diverse natural images that capture the high-level structure of the training data. While such unconditional models are fascinating in their own right, many of the practical applications of image modelling require the model to be conditioned on prior information: for example, an image model used for reinforcement learning planning in a visual environment would need to predict future scenes given specific states and actions [17]. Similarly image processing tasks such as denoising, deblurring, inpainting, super-resolution and colorization rely on generating improved images conditioned on noisy or incomplete data. Neural artwork [18, 5] and content generation represent potential future uses for conditional generation.

This paper explores the potential for conditional image modelling by adapting and improving a convolutional variant of the PixelRNN architecture [30]. As well as providing excellent samples, this network has the advantage of returning explicit probability densities (unlike alternatives such as generative adversarial networks [6, 3, 19]), making it straightforward to apply in domains such as compression [32] and probabilistic planning and exploration [2]. The basic idea of the architecture is to use autoregressive connections to model images pixel by pixel, decomposing the joint image distribution as a product of conditionals. Two variants were proposed in the original paper: PixelRNN, where the pixel distributions are modeled with two-dimensional LSTM [7, 26], and PixelCNN, where they are modelled with convolutional networks. PixelRNNs generally give better performance, but PixelCNNs are much faster to train because convolutions are inherently easier to parallelize; given

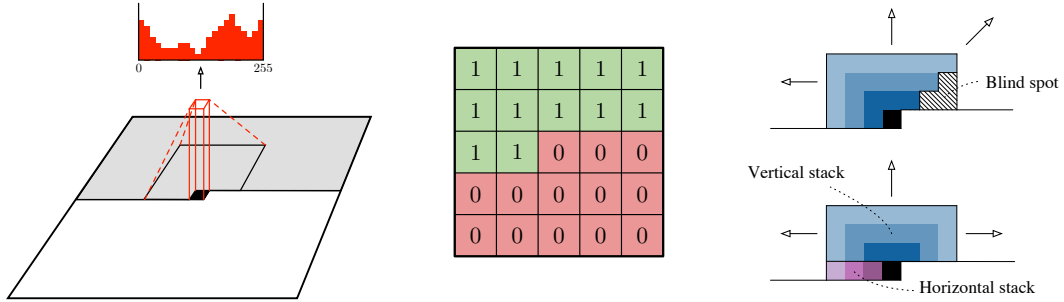

Figure 1: **Left**: A visualization of the PixelCNN that maps a neighborhood of pixels to prediction for the next pixel. To generate pixel $x_i$ the model can only condition on the previously generated pixels $x_1, \ldots x_{i-1}$. **Middle**: an example matrix that is used to mask the 5x5 filters to make sure the model cannot read pixels below (or strictly to the right) of the current pixel to make its predictions. **Right**: Top: PixelCNNs have a *blind spot* in the receptive field that can not be used to make predictions. Bottom: Two convolutional stacks (blue and purple) allow to capture the whole receptive field.

the vast number of pixels present in large image datasets this is an important advantage. We aim to combine the strengths of both models by introducing a gated variant of PixelCNN (Gated PixelCNN) that matches the log-likelihood of PixelRNN on both CIFAR and ImageNet, while requiring less than half the training time.

We also introduce a conditional variant of the Gated PixelCNN (Conditional PixelCNN) that allows us to model the complex conditional distributions of natural images given a latent vector embedding. We show that a single Conditional PixelCNN model can be used to generate images from diverse classes such as dogs, lawn mowers and coral reefs, by simply conditioning on a one-hot encoding of the class. Similarly one can use embeddings that capture high level information of an image to generate a large variety of images with similar features. This gives us insight into the invariances encoded in the embeddings — e.g., we can generate different poses of the same person based on a single image. The same framework can also be used to analyse and interpret different layers and activations in deep neural networks.

## 2 Gated PixelCNN

PixelCNNs (and PixelRNNs) [30] model the joint distribution of pixels over an image $\mathbf{x}$ as the following product of conditional distributions, where $x_i$ is a single pixel:

$$p(\mathbf{x}) = \prod_{i=1}^{n^2} p(x_i | x_1, ..., x_{i-1}). \qquad (1)$$

The ordering of the pixel dependencies is in raster scan order: row by row and pixel by pixel within every row. Every pixel therefore depends on all the pixels above and to the left of it, and not on any of other pixels. The dependency field of a pixel is visualized in Figure 1 (left).

A similar setup has been used by other autoregressive models such as NADE [14] and RIDE [26]. The difference lies in the way the conditional distributions $p(x_i | x_1, ..., x_{i-1})$ are constructed. In PixelCNN every conditional distribution is modelled by a convolutional neural network. To make sure the CNN can only use information about pixels above and to the left of the current pixel, the filters of the convolution are *masked* as shown in Figure 1 (middle). For each pixel the three colour channels (R, G, B) are modelled successively, with B conditioned on (R, G), and G conditioned on R. This is achieved by splitting the feature maps at every layer of the network into three and adjusting the centre values of the mask tensors. The 256 possible values for each colour channel are then modelled using a softmax.

PixelCNN typically consists of a stack of masked convolutional layers that takes an N x N x 3 image as input and produces N x N x 3 x 256 predictions as output. The use of convolutions allows the predictions for all the pixels to be made in parallel during training (all conditional distributions from

Equation 1). During sampling the predictions are sequential: every time a pixel is predicted, it is fed back into the network to predict the next pixel. This sequentiality is essential to generating high quality images, as it allows every pixel to depend in a highly non-linear and multimodal way on the previous pixels.

## 2.1 Gated Convolutional Layers

PixelRNNs, which use spatial LSTM layers instead of convolutional stacks, have previously been shown to outperform PixelCNNs as generative models [30]. One possible reason for the advantage is that the recurrent connections in LSTM allow every layer in the network to access the entire neighbourhood of previous pixels, while the region of the neighbourhood available to pixelCNN grows linearly with the depth of the convolutional stack. However this shortcoming can largely be alleviated by using sufficiently many layers. Another potential advantage is that PixelRNNs contain multiplicative units (in the form of the LSTM gates), which may help it to model more complex interactions. To amend this we replaced the rectified linear units between the masked convolutions in the original pixelCNN with the following gated activation unit:

$$\mathbf{y} = \tanh(W_{k,f} * \mathbf{x}) \odot \sigma(W_{k,g} * \mathbf{x}), \tag{2}$$

where $\sigma$ is the sigmoid non-linearity, $k$ is the number of the layer, $\odot$ is the element-wise product and $*$ is the convolution operator. We call the resulting model the Gated PixelCNN. Feed-forward neural networks with gates have been explored in previous works, such as highway networks [25], grid LSTM [13] and neural GPUs [12], and have generally proved beneficial to performance.

## 2.2 Blind spot in the receptive field

In Figure 1 (top right), we show the progressive growth of the effective receptive field of a $3 \times 3$ masked filter over the input image. Note that a significant portion of the input image is ignored by the masked convolutional architecture. This 'blind spot' can cover as much as a quarter of the potential receptive field (e.g., when using 3x3 filters), meaning that none of the content to the right of the current pixel would be taken into account.

In this work, we remove the blind spot by combining two convolutional network stacks: one that conditions on the current row so far (horizontal stack) and one that conditions on all rows above (vertical stack). The arrangement is illustrated in Figure 1 (bottom right). The vertical stack, which does not have any masking, allows the receptive field to grow in a rectangular fashion without any blind spot, and we combine the outputs of the two stacks after each layer. Every layer in the horizontal stack takes as input the output of the previous layer as well as that of the vertical stack. If we had connected the output of the horizontal stack into the vertical stack, it would be able to use information about pixels that are below or to the right of the current pixel which would break the conditional distribution.

Figure 2 shows a single layer block of a Gated PixelCNN. We combine $W_f$ and $W_g$ in a single (masked) convolution to increase parallelization. As proposed in [30] we also use a residual connection [11] in the horizontal stack. We have experimented with adding a residual connection in the vertical stack, but omitted it from the final model as it did not improve the results in our initial experiments. Note that the $(n \times 1)$ and $(n \times n)$ masked convolutions in Figure 2 can also be implemented by $(\lceil \frac{n}{2} \rceil \times 1)$ and $(\lceil \frac{n}{2} \rceil \times n)$ convolutions followed by a shift in pixels by padding and cropping.

## 2.3 Conditional PixelCNN

Given a high-level image description represented as a latent vector $\mathbf{h}$, we seek to model the conditional distribution $p(\mathbf{x}|\mathbf{h})$ of images suiting this description. Formally the conditional PixelCNN models the following distribution:

$$p(\mathbf{x}|\mathbf{h}) = \prod_{i=1}^{n^2} p(x_i|x_1, ..., x_{i-1}, \mathbf{h}). \tag{3}$$

We model the conditional distribution by adding terms that depend on $\mathbf{h}$ to the activations before the nonlinearities in Equation 2, which now becomes:

$$\mathbf{y} = \tanh(W_{k,f} * \mathbf{x} + V_{k,f}^T \mathbf{h}) \odot \sigma(W_{k,g} * \mathbf{x} + V_{k,g}^T \mathbf{h}), \tag{4}$$

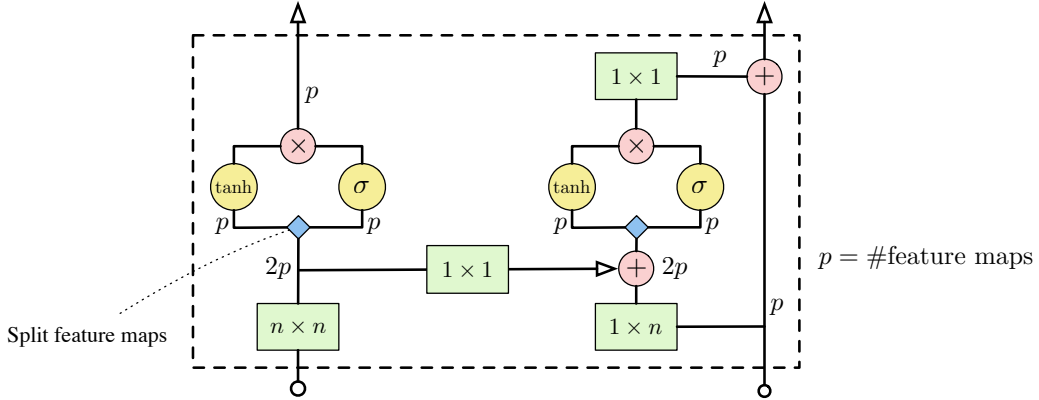

Figure 2: A single layer in the Gated PixelCNN architecture. Convolution operations are shown in green, element-wise multiplications and additions are shown in red. The convolutions with $W_f$ and $W_g$ from Equation 2 are combined into a single operation shown in blue, which splits the $2p$ features maps into two groups of $p$.

where $k$ is the layer number. If $\mathbf{h}$ is a one-hot encoding that specifies a class this is equivalent to adding a class dependent bias at every layer. Notice that the conditioning does not depend on the location of the pixel in the image; this is appropriate as long as $\mathbf{h}$ only contains information about *what* should be in the image and not *where*. For example we could specify that a certain animal or object should appear, but may do so in different positions and poses and with different backgrounds.

We also developed a variant where the conditioning function was location dependent. This could be useful for applications where we do have information about the location of certain structures in the image embedded in $\mathbf{h}$. By mapping $\mathbf{h}$ to a spatial representation $\mathbf{s} = m(\mathbf{h})$ (which has the same width and height as the image but may have an arbitrary number of feature maps) with a deconvolutional neural network $m()$, we obtain a location dependent bias as follows:

$$\mathbf{y} = \tanh(W_{k,f} * \mathbf{x} + V_{k,f} * \mathbf{s}) \odot \sigma(W_{k,g} * \mathbf{x} + V_{k,g} * \mathbf{s}). \tag{5}$$

where $V_{k,g} * \mathbf{s}$ is an unmasked $1 \times 1$ convolution.

### 2.4 PixelCNN Auto-Encoders

Because conditional PixelCNNs have the capacity to model diverse, multimodal image distributions $p(\mathbf{x}|\mathbf{h})$, it is possible to apply them as image decoders in existing neural architectures such as auto-encoders. An auto-encoder consists of two parts: an encoder that takes an input image $\mathbf{x}$ and maps it to a (usually) low-dimensional representation $\mathbf{h}$, and a decoder that tries to reconstruct the original image.

Starting with a traditional convolutional auto-encoder architecture [16], we replace the deconvolutional decoder with a conditional PixelCNN and train the complete network end-to-end. Since PixelCNN has proved to be a strong unconditional generative model, we would expect this change to improve the reconstructions. Perhaps more interestingly, we also expect it to change the representations that the encoder will learn to extract from the data: since so much of the low level pixel statistics can be handled by the PixelCNN, the encoder should be able to omit these from $\mathbf{h}$ and concentrate instead on more high-level abstract information.

## 3 Experiments

### 3.1 Unconditional Modeling with Gated PixelCNN

Table 1 compares Gated PixelCNN with published results on the CIFAR-10 dataset. These architectures were all optimized for the best possible validation score, meaning that models that get a lower

score actually generalize better. Gated PixelCNN outperforms the PixelCNN by 0.11 *bits/dim*, which has a very significant effect on the visual quality of the samples produced, and which is close to the performance of PixelRNN.

| Model | NLL Test (Train) |
|---|---|
| Uniform Distribution: [30] | 8.00 |
| Multivariate Gaussian: [30] | 4.70 |
| NICE: [4] | 4.48 |
| Deep Diffusion: [24] | 4.20 |
| DRAW: [9] | 4.13 |
| Deep GMMs: [31, 29] | 4.00 |
| Conv DRAW: [8] | 3.58 (3.57) |
| RIDE: [26, 30] | 3.47 |
| PixelCNN: [30] | 3.14 (3.08) |
| PixelRNN: [30] | 3.00 (2.93) |
| **Gated PixelCNN**: | 3.03 (2.90) |

Table 1: Test set performance of different models on CIFAR-10 in *bits/dim* (lower is better), training performance in brackets.

In Table 2 we compare the performance of Gated PixelCNN with other models on the ImageNet dataset. Here Gated PixelCNN outperforms PixelRNN; we believe this is because the models are underfitting, larger models perform better and the simpler PixelCNN model scales better. We were able to achieve similar performance to the PixelRNN (Row LSTM [30]) in less than half the training time (60 hours using 32 GPUs). For the results in Table 2 we trained a larger model with 20 layers (Figure 2), each having 384 hidden units and filter size of $5 \times 5$. We used $200K$ synchronous updates over 32 GPUs in TensorFlow [1] using a total batch size of 128.

| 32x32 | Model | NLL Test (Train) |
|---|---|---|
| | Conv Draw: [8] | 4.40 (4.35) |
| | PixelRNN: [30] | 3.86 (3.83) |
| | **Gated PixelCNN**: | 3.83 (3.77) |
| **64x64** | **Model** | **NLL Test (Train)** |
| | Conv Draw: [8] | 4.10 (4.04) |
| | PixelRNN: [30] | 3.63 (3.57) |
| | **Gated PixelCNN**: | 3.57 (3.48) |

Table 2: Performance of different models on ImageNet in *bits/dim* (lower is better), training performance in brackets.

## 3.2 Conditioning on ImageNet Classes

For our second experiment we explore class-conditional modelling of ImageNet images using Gated PixelCNNs. Given a one-hot encoding $\mathbf{h}_i$ for the $i$-th class we model $p(\mathbf{x}|\mathbf{h}_i)$. The amount of information that the model receives is only $\log(1000) \approx 0.003$ bits/pixel (for a 32x32 image). Still, one could expect that conditioning the image generation on class label could significantly improve the log-likelihood results, however we did not observe big differences. On the other hand, as noted in [27], we observed great improvements in the visual quality of the generated samples.

In Figure 3 we show samples from a single class-conditional model for 8 different classes. We see that the generated classes are very distinct from one another, and that the corresponding objects, animals and backgrounds are clearly produced. Furthermore the images of a single class are very diverse: for example the model was able to generate similar scenes from different angles and lightning conditions. It is encouraging to see that given roughly 1000 images from every animal or object the model is able to generalize and produce new renderings.

### 3.3 Conditioning on Portrait Embeddings

In our next experiment we took the latent representations from the top layer of a convolutional network trained on a large database of portraits automatically cropped from Flickr images using a face detector. The quality of images varied wildly, because a lot of the pictures were taken with mobile phones in bad lightning conditions.

The network was trained with a triplet loss function [23] that ensured that the embedding $\mathbf{h}$ produced for an image $\mathbf{x}$ of a specific person was closer to the embeddings for all other images of the same person than it was to any embedding of another person.

After the supervised net was trained we took the (image=$\mathbf{x}$, embedding=$\mathbf{h}$) tuples and trained the Conditional PixelCNN to model $p(\mathbf{x}|\mathbf{h})$. Given a new image of a person that was not in the training set we can compute $\mathbf{h} = f(\mathbf{x})$ and generate new portraits of the same person.

Samples from the model are shown in Figure 4. We can see that the embeddings capture a lot of the facial features of the source image and the generative model is able to produce a large variety of new faces with these features in new poses, lighting conditions, etc.

Finally, we experimented with reconstructions conditioned on linear interpolations between embeddings of pairs of images. The results are shown in Figure 5. Every image in a single row used the same random seed in the sampling which results in smooth transitions. The leftmost and rightmost images are used to produce the end points of interpolation.

### 3.4 PixelCNN Auto Encoder

This experiment explores the possibility of training both the encoder and decoder (PixelCNN) end-to-end as an auto-encoder. We trained a PixelCNN auto-encoder on 32x32 ImageNet patches and compared the results with those from a convolutional auto-encoder trained to optimize MSE. Both models used a 10 or 100 dimensional bottleneck.

Figure 6 shows the reconstructions from both models. For the PixelCNN we sample multiple conditional reconstructions. These images support our prediction in Section 2.4 that the information encoded in the bottleneck representation $\mathbf{h}$ will be qualitatively different with a PixelCNN decoder than with a more conventional decoder. For example, in the lowest row we can see that the model generates different but similar looking indoor scenes with people, instead of trying to exactly reconstruct the input.

## 4 Conclusion

This work introduced the Gated PixelCNN, an improvement over the original PixelCNN that is able to match or outperform PixelRNN [30], and is computationally more efficient. In our new architecture, we use two stacks of CNNs to deal with "blind spots" in the receptive field, which limited the original PixelCNN. Additionally, we use a gating mechanism which improves performance and convergence speed. We have shown that the architecture gets similar performance to PixelRNN on CIFAR-10 and is now state-of-the-art on the ImageNet 32x32 and 64x64 datasets.

Furthermore, using the Conditional PixelCNN we explored the conditional modelling of natural images in three different settings. In class-conditional generation we showed that a single model is able to generate diverse and realistic looking images corresponding to different classes. On human portraits the model is capable of generating new images from the same person in different poses and lightning conditions from a single image. Finally, we demonstrated that the PixelCNN can be used as a powerful image decoder in an autoencoder. In addition to achieving state of the art log-likelihood scores in all these datasets, the samples generated from our model are of very high visual quality showing that the model captures natural variations of objects and lighting conditions.

In the future it might be interesting to try and generate new images with a certain animal or object solely from a single example image [21, 22]. Another exciting direction would be to combine Conditional PixelCNNs with variational inference to create a variational auto-encoder. In existing work $p(\mathbf{x}|\mathbf{h})$ is typically modelled with a Gaussian with diagonal covariance and using a PixelCNN instead could thus improve the decoder in VAEs. Another promising direction of this work would be to model images based on an image caption instead of class label [15, 19].

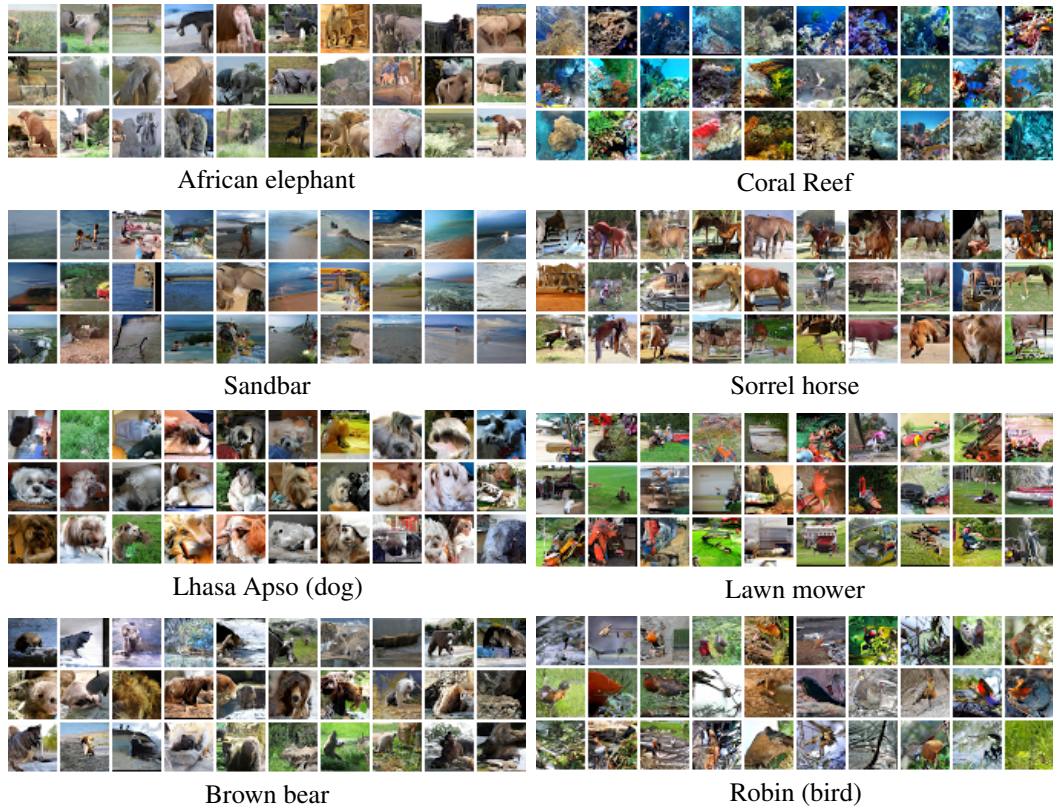

African elephant

Coral Reef

Sandbar

Sorrel horse

Lhasa Apso (dog)

Lawn mower

Brown bear

Robin (bird)

Figure 3: Class-Conditional samples from the Conditional PixelCNN.

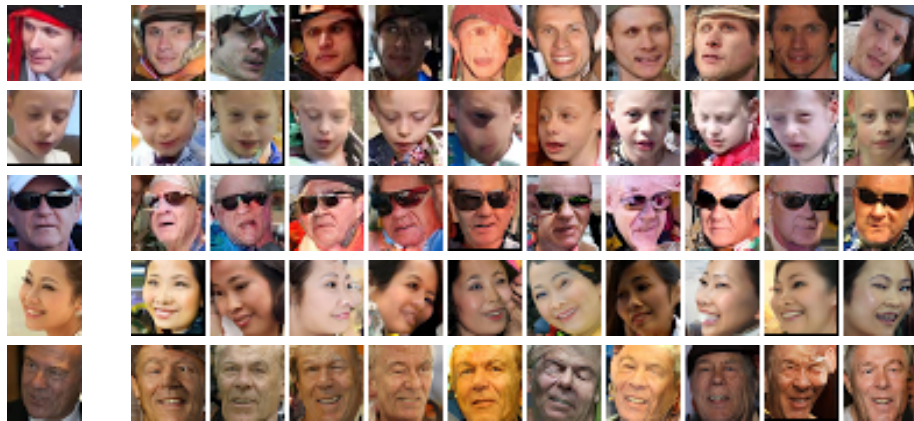

Figure 4: **Left**: source image. **Right**: new portraits generated from high-level latent representation.

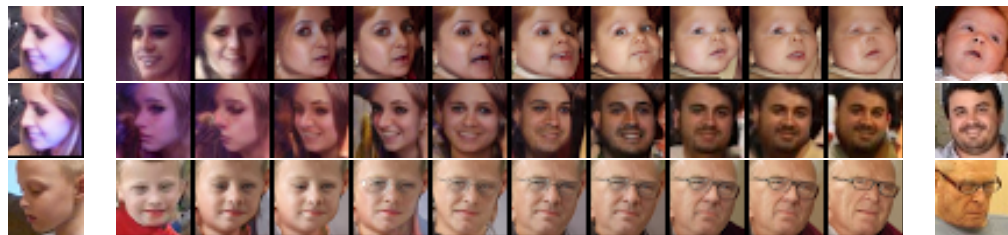

Figure 5: Linear interpolations in the embedding space decoded by the PixelCNN. Embeddings from leftmost and rightmost images are used for endpoints of the interpolation.

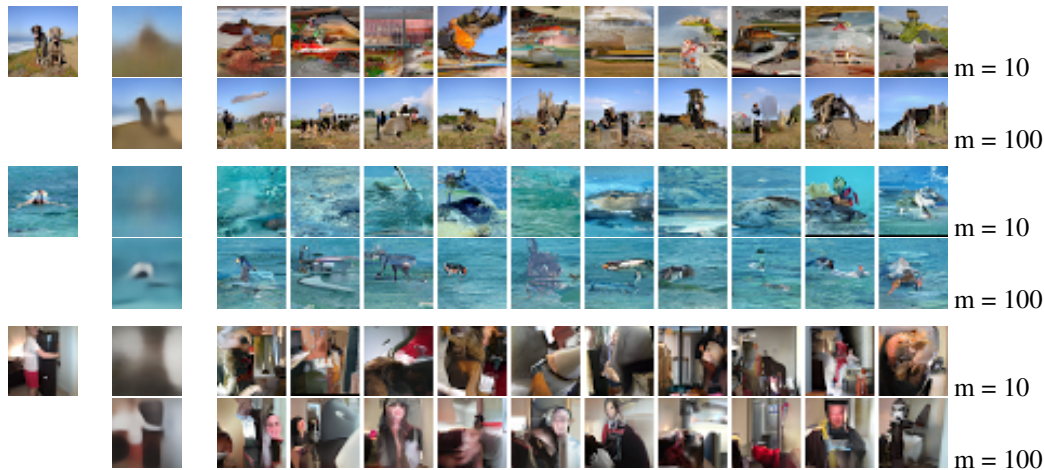

m = 10

m = 100

m = 10

m = 100

m = 10

m = 100

Figure 6: Left to right: original image, reconstruction by an auto-encoder trained with MSE, conditional samples from a PixelCNN auto-encoder. Both auto-encoders were trained end-to-end with a $m = 10$-dimensional bottleneck and a $m = 100$ dimensional bottleneck.

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
