[Supplementary Material]

# Conditional Image Generation with PixelCNN Decoders

**Aäron van den Oord**
Google DeepMind
avdnoord@google.com

**Nal Kalchbrenner**
Google DeepMind
nalk@google.com

**Oriol Vinyals**
Google DeepMind
vinyals@google.com

**Lasse Espeholt**
Google DeepMind
espeholt@google.com

**Alex Graves**
Google DeepMind
gravesa@google.com

**Koray Kavukcuoglu**
Google DeepMind
korayk@google.com

## Appendix

Geyser

Hartebeest

Grey whale

Tiger

EntleBucher (dog)

Yellow lady's slipper (flower)

Figure 1: Class-Conditional (multi-scale $64 \times 64$) samples from the Conditional Pixel CNN.

African elephant

Coral Reef

Sandbar

Sorrel horse

Lhasa Apso (dog)

Lawn mower

Brown bear

Robin (bird)

Figure 2: Class-Conditional $32 \times 32$ samples from the Conditional Pixel CNN.

Figure 3: **Left**: source image. **Right**: new portraits generated from high-level latent representation.

Figure 4: Linear interpolations in the embedding space decoded by the Pixel CNN. Embeddings from leftmost and rightmost images are used for endpoints of the interpolation.

Figure 5: Left to right: original image, reconstruction by an auto-encoder trained with MSE, conditional samples from a Pixel CNN auto-encoder. Both auto-encoders were trained end-to-end with a $m = 10$-dimensional bottleneck and a $m = 100$ dimensional bottleneck.