[Reviews · NeurIPS 2016]

Reviewer 1

Summary

This paper extends the pixel CNN model of Oord et al. by adding a gating function to the nonlinearities and by conditioning on additional information. It also proposes a two-stack CNN to deal with the blind spot problem of the original model when using masked convolutions. The experiments are compelling, showing a nice improvement over the pixel CNN with performance matching or exceeding the pixel RNN. The ability to condition on additional information is explored in various interesting ways, from the obvious condition on a class idea, to generating novel views of a face given an example input, to a generative auto-encoder that optimizes log-likelihood rather than MSE.

Qualitative Assessment

This is an excellent paper that I think certainly deserves acceptance. In particular, the generated images are among the most interesting and high-quality that I have seen so far. The model is elegant, the experiments are convincing, and the paper is written extremely well. The most significant criticism I have is that there are several new pieces and it's not entirely clear where the win is coming from. Namely, the introduction of gating nonlinearities, the two-stack architecture, and the residual connections in the horizontal stack. Presumably combining these ingredients led to the best results in the end, but it would be nice to gauge the relative contribution of each piece. One thing that wasn't entirely clear is how to deal with image boundaries. I'm guessing that the images are zero-padded before applying the CNN? If so, this should be stated clearly. I'm guessing that the spatial representation in section 2.3, discussing the deconvolutional m(h), is used in the auto-encoding experiments? If this is the case then I think this connection should be made clear. It would be nice to have more details on the convolutional auto-encoder baseline in section 3.4. Either a reference, or a quick description in the appendix.

Confidence in this Review

3-Expert (read the paper in detail, know the area, quite certain of my opinion)


Reviewer 2

Summary

This paper investigates the problem of conditional image generation based on the pixel (R/C)NN framework. Building upon the previous pixel (R/C)NN framework, this paper proposes a gated extension of pixel CNN using multiplicative interactions which can be trained efficiently. The main focus of this paper lies in the conditioning modeling of image distribution p(x|h), where h is assumed to be a vector (e.g., one-hot vector or feature vector from pre-trained encoder networks). In addition, the proposed gated pixel CNN can be jointly trained with an encoder network, which is referred as pixel CNN auto-encoders. The model is validated in experiments on CIFAR-10, ImageNet, and Portraits dataset. For unconditional modeling, the performance of gated extension is on par with (slightly better on ImageNet) original pixel CNN. For conditional modeling, the model is able to generate reasonably-looking and diverse samples that agrees with the conditioning vector (either class label or face embedding).

Qualitative Assessment

The paper solves a significant problem in generative modeling and the paper is quite interesting. However, reviewer feels the current version is not polished well due to several issues in the experimental section. For rebuttal, please focus on the (*), (**), (***) and (***) mentioned in the following paragraphs. Reviewer is willing to change the score if all the concerns are addressed in the rebuttal. Novelty: The proposed model is technically novel in the sense that it explores the conditional modeling with the recent pixel (R/C)NN framework. The extension of pixel-CNN to auto-encoders is interesting. In a high-level, however, generating images with conditioning variables has been explored to some extent: -- Learning to Generate Chairs with Convolutional Neural Networks by Dosovitskiy et al. -- Deep Generative Image Modeling using a Laplacian Pyramid of Adversarial Networks by Denton et al. -- Deep Convolutional Inverse Graphics Network by Kulkarni et al. -- Attribute2Image: Conditional Image Generation from Visual Attributes by Yan et al. -- Generating Images from Captions with Attention by Mansimov et al. Clarity: The paper is easy to follow and clearly written in general. However, it is hard to reproduce the results solely based on the description. Reviewer suggests the authors to submit supplementary document or consider code release in the future. References are incomplete. Technical quality: The current experimental result fails to convince reviewer for the following reasons: -- In unconditional modeling, whether the proposed gated pixel CNN is indeed significant enough compared to original pixel CNN [21]? In the paper, the only comparisons between original pixel CNN and gated pixel CNN are detailed in Table 1 and Table 2. However, it is possible that performance gap comes from more parameters and deeper architecture of gated pixel CNN. *For rebuttal, please describe the quality of samples (comment the difference compared to original pixel CNN) and report the numbers under fair comparison. -- In conditional modeling, whether the improvements in visual quality is a sign of overfitting? It would be more convincing if the nearest training examples are added along with generated samples (as has been done in LAP-GAN paper). **For rebuttal, please comment on the overfitting risk and visual similarity between generated samples and nearest training examples. -- In conditional modeling, whether the proposed model is superior to original pixel CNN and other existing previous work is unknown. It seems that Eq.(4) and Eq.(5) can also be applied to original pixel CNN [21] as well. For other existing work on conditional image generation, please refer to class-conditional generation on CIFAR-10 dataset (LAP-GAN) and text-conditional generation on MS-COCO dataset (alignDRAW [10]). It would be more convincing if the proposed model is evaluated against the original pixel CNN [21] or other existing previous work. ***For rebuttal, please comment on the performance of conditional generation using original pixel CNN. Usefulness: One issue with the original pixel (R/C)NN framework is that the generation cannot be made in parallel (one needs to generate one pixel at a time). This would be an issue in practice since the competitor models (condVAE and condGAN) can generate images in a much more efficient way. ****For rebuttal, please make it clear if this problem has been resolved in the proposed method. Detailed Comments: -- Eq. (2): “W_{g,f}” → “W_{k,g}” -- Eq. (4) vs. Eq. (5): are you applying conditioning variable to every pixel CNN layer or not? Which one has been used in the experiments (imagenet and portraits)? Any suggestions when choosing Eq. (4) or Eq. (5) in practice? -- The description of the “portraits” dataset that being used is missing. How many images are there in the dataset? Whether the identities are overlapping between training set and testing set?

Confidence in this Review

3-Expert (read the paper in detail, know the area, quite certain of my opinion)


Reviewer 3

Summary

Modelling the distribution of natural images is a longstanding problem with recent rapid progress made by borrowing from deep learning, e.g. the pixel RNN and pixel CNN (Oord et al 2016). This paper presents a more computationally efficient alternative, the gated pixel CNN, and extends the work by by conditioning on labels by using a class dependent bias.

Qualitative Assessment

This work is largely derivative of the Pixel RNN work, and without the conditional model there would not be sufficient novelty. This is especially considering that while setting a new state of the art, the actual log-likelihood performance difference to Pixel RNN is rather miniscule. But in addition to the class-conditional model that the authors note as the main point of novelty, posing the model as and encoder-decoder opens a lot of interesting possibilities. The authors explore this e.g. by performing manipulations to the latent representation in the face-morphing example. Both of these lines of work have the potential for impacting a number of applications, and have not been similarly investigated in previous work. In my view this is a more important contribution than the proposed gated pixel CNN. Even thought the 2x speedup over the RNN is impressive, it will probably not be the deciding factor in picking a natural image model for an application. Misc small points: W_f and W_g are not defined. Neither is V. Check equation (3), which seems to imply p(x) = p(x,h) Explain how dimensionalities fit together in equation (4). Is y a feature map or a single pixel? If the former, how does the output V^T h map to it? Why the change of notation from the scalar y in (4) to the bold y_{i,j} in (5)? Would like to see pixel CNN likelihood numbers for ImageNet in table 2. Line 207 seems to be an unfinished sentence. Please provide more details on the model used for the face encoder, e.g. dimensionality of the hidden state, and source of the dataset if it is public.

Confidence in this Review

3-Expert (read the paper in detail, know the area, quite certain of my opinion)


Reviewer 4

Summary

This paper extends the PixelCNN architecture from van den Ord et al. (ICML 2016), and provides various supporting experiments.

Qualitative Assessment

This paper presents improvements to the PixelCNN architecture from the earlier PixelRNN paper (ICML 2016). The proposed changes are fairly minor, but seem beneficial empirically. The production quality of the paper is high and the technical descriptions are clear. The quantitative and qualitative results are impressive, though I don't know how beneficial it will be to most researchers that this approach reduces training time to only 60 hours on a 32 GPU cluster. It would be nice if the authors could describe how to efficiently sample from this architecture, if it is possible to do so. Based on my understanding of the model, this seems pretty annoying to implement without wasting a lot of computation. The resized ImageNet data should be made publicly available. The ability of topologically-informed autoregressive models to precisely capture local dynamics will probably be more useful when combined with techniques (e.g. latent variables and SGVB) that efficiently capture global structure. The slow growth in effective receptive field for each generated pixel seems like a significant hindrance in the PixelCNN's current form.

Confidence in this Review

2-Confident (read it all; understood it all reasonably well)


Reviewer 5

Summary

The paper addresses the problem of image generation and continues in the direction of the PixelCNN, introduced in prior work. The contribution of the paper consists of three parts: 1. By proposing a new gated architecture, the paper improves the quality of PixelCNN to the level close to PixelRNN, while being 2 times faster in the training time than PixelRNN. The claim is justified by experiments on CIFAR10 and Imagenet. 2. The paper proposes a conditioning scheme which allows to generate images related to conditioning information. Conditioning, explored in the experiment: a) image class for general images b) separately trained face embedding 3. The paper demonstrates experimentally that the proposed PixelCNN architecture can be used as a decoder in autoencoders.

Qualitative Assessment

Overall, the paper is well-written and serves as a decent continuation of the PixelCNN/RNN work. Below I give my comments on how it could be improved and pose some questions for the authors. lines 14, 226: There are multiple instances of claims in the paper that the proposed method "significantly reduces computational requirements". While this is true for the training time, the generation time remains the same because pixels need to be generated in order. The parts of the text mentioned above should be corrected by explicitly saying it's the training time that improved. line 33: Audio example is not justified by experiments and should be removed. line 87: "to" missing section 2.2: writing should be improved. It ambiguous how the vertical stack works and figure 1 is confusing. Does the vertical stack mean the line segment above? How does it grow to the sides then? Otherwise, if it also includes the pixels on the sides, it would also cover pixels to the right of the predicted one. Please, clarify this in the writing and figure 1. In the figure 2, what is n? Is it image size? Since you say convolution is in green, that would imply you use convolution image_size x image_size which is probably not what you do. Finally, it was not clear to me how the vertical and horizontal stacks are combined, please try to make it more clear in writing. line 129: term V should be introduced before using it (in fact, it's not introduced anywhere in the paper). Is it a fixed coefficient matrix or a learned one? I assume for conditioning experiment with classes it was just identity matrix and for experiment with supervised representation is was learned. If so, it should be specified in the respective experimental sections, what V is used in each. line 166: the claim that 0.11 bits/dim difference has a very significant effect on visual quality should either be justified or removed from the manuscript. line 170: what does "models are in underfitted regime" mean? line 173: when a larger model with 20 layers is used, what happens to models from previous work (pixelCNN/RNN)? Are they also enlarged proportionally? If not, additional ablation experiment would be required to understand if proposed "gating" for CNN really outperforms pixelRNN or that is the result of simply using larger model. line 177: as mentioned earlier, V should be clarified section 3.3: general question: Isn't it possible to do the same conditional generation with pixelRNN? It means compute hidden h for input sample, then fix this hidden state and generate x? Isn't it a better and more natural solution? Have you tried that? In figure 4, column 5 I noticed that some humans do not have 2 eyes, only 1, although all training data had 2 eyes, I assume. Do you think it could be consequence of separately training face representation? I can imagine knowing that there are 2 eyes is not a discriminative feature for training with triplet loss as it doesn't help to tell if the person is different or not. Do you think using pixelRNN the way I described above would help to resolve this problem?

Confidence in this Review

2-Confident (read it all; understood it all reasonably well)


Reviewer 6

Summary

In this paper, based on the recent PixelCNN algorithm, the authors proposed the conditional PixelCNN to generate a large diversity of realistic images conditioned on a given class label or latent vector. Actually, the authors combined both advantages of PixelCNN and PixelRNN, where the gate mechanism in PixelRNN is introduced to PixelCNN, and therefore the proposed algorithm can achieve the similar performance as PixelRNN in much less time than PixelCNN.

Qualitative Assessment

I like the nice idea presented in this paper and I'm also deeply impressive by the results in the experimental part. The combination of RNN and CNN will be a good inspiration in this field. On the other hand, although the generated images are really amazing when we take a quick glance at them, they seem weird, even a little creepy when we carefully look at them, especially for those class-conditional samples. Maybe it would be better to put more constrains on the given class label or the loss.

Confidence in this Review

3-Expert (read the paper in detail, know the area, quite certain of my opinion)